# Agreement in the Postural Assessment of Older Adults by Physical Therapists Using Clinical and Imaging Methods

**DOI:** 10.3390/geriatrics9020040

**Published:** 2024-03-22

**Authors:** Naoki Sugiyama, Yoshihiro Kai, Hitoshi Koda, Toru Morihara, Noriyuki Kida

**Affiliations:** 1Department of Advanced Fibro-Science, Kyoto Institute of Technology, Hashikami-cho, Matsugasaki, Sakyo-ku, Kyoto 606-8585, Japan; nsugiyama.u@gmail.com; 2Department of Physical Therapy, Faculty of Health Sciences, Kyoto Tachibana University, 34 Yamada-cho, Oyake, Yamashina-ku, Kyoto 607-8175, Japan; kai-y@tachibana-u.ac.jp; 3Department of Rehabilitation Sciences, Faculty of Allied Health Sciences, Kansai University of Welfare Sciences, Asahigaoka 3-11-1, Kashiwara-shi 582-0026, Japan; h-koda@tamateyama.ac.jp; 4Marutamachi Rehabilitation Clinic, Nishinokyo Kurumazakacho Nakagyo-ku, Kyoto 604-8405, Japan; toru4271@koto.kpu-m.ac.jp; 5Faculty of Arts and Sciences, Kyoto Institute of Technology, Hashikami-cho, Matsugasaki, Sakyo-ku, Kyoto 606-8585, Japan

**Keywords:** posture assessment, imaging method, years of experience

## Abstract

Postural assessment is one of the indicators of health status in older adults. Since the number of older adults is on the rise, it is essential to assess simpler methods and automated ones in the future. Therefore, we focused on a visual method (imaging method). The purpose of this study is to determine the degree of agreement between the imaging method and the palpation and visual methods (clinical method). In addition, the influence of differences in the information content of the sagittal plane images on the assessment was also investigated. In this experiment, 28 sagittal photographs of older adults whose posture had already been assessed using the clinical method were used. Furthermore, based on these photographs, 28 gray and silhouette images (G and S images) were generated, respectively. The G and S images were assessed by 28 physical therapists (PTs) using the imaging method. The assessment was based on the Kendall classification, with one of four categories selected for each image: ideal, kyphosis lordosis, sway back, and flat back. Cross-tabulation matrices of the assessments using the clinical method and imaging method were created. In this table, four categories and two categories of ideal and non-ideal (KL, SB, and FB) were created. The agreement was evaluated using the prevalence-adjusted bias-adjusted kappa (PABAK). In addition, sensitivity and specificity were calculated to confirm the reliability. When comparing the clinical and imaging methods in the four posture categories, the PABAK values were −0.14 and −0.29 for the S and G images, respectively. In the case of the two categories, the PABAK values were 0.57 and 0.5 for the S and G images, respectively. The sensitivity and specificity were 86% and 57% for the S images and 76% and 71% for the G images, respectively. The four categories show that the imaging method is difficult to assess regardless of the image processing. However, in the case of the two categories, the same assessment of the clinical method applied to the imaging method for both the S and G images. Therefore, no differences in image processing were observed, suggesting that PTs can identify posture using the visual method.

## 1. Introduction

Japan has the highest number of older adults worldwide and the longest lifespan [1], accounting for 21% of its population [2]. Consequently, the Ministry of Health has identified health-adjusted life expectancy (HALE) as an important problem in Japan [3]. Health-adjusted life expectancy is defined as “the period of time during which a person can live without being limited in daily life by health problems” [4]. The time between HALE and lifespan is related to a period of poor health, during which daily life is restricted. A typical example of poor health is being bedridden, which causes not only physical but also emotional and social problems [5]. Being bedridden for 10 days results in a significant decrease in physical activity [6]. Therefore, the prevention of bedridden status plays an important role in the development of HALE. One way to prevent being bedridden, as recommended by the Ministry of Health, Labor, and Welfare [7], is to maintain and enhance the muscle strength of patients through methods that can be easily incorporated into daily life. This further decreases the risk of falls, which is a factor that is associated with being bedridden [8]. Falls occur when one’s balance is disturbed beyond the ability to control one’s posture [9], which is considered an indicator of muscle strength in older adults. According to Nishiwaki et al., poor posture is associated with an increased risk of bone fractures [10,11]. Furthermore, posture is associated with mental states, and flexion may occur in negative states, such as anxiety [12,13,14]. Thus, posture assessment can be used for the medical inspection of a broad spectrum of mental conditions, as well as physical health.

Physical assessment is generally performed in posture assessments using visual and palpation methods by physical therapists (PTs). Numerous assessment methods exist, depending on the definition of an ideal posture. Among them is the Kendall classification, which is a posture assessment tool widely used worldwide [15]. The Kendall classification is divided into four categories: ideal posture (ideal), which is a state of low strain on the muscles and bones; kyphosis lordosis (KL) characterized by hyperkyphosis in the thoracic spine, hyperlordosis in the lumbar spine, and anterior tilt of the pelvis; sway back (SB), characterized by hyperkyphosis in the thoracic spine, flattening of the lumbar spine, and posterior tilt of the pelvis; and flat back (FB), characterized by hyperkyphosis in the upper thoracic spine, flattening of the lower thoracic and lumbar spines, and posterior tilt of the pelvis. Four postures were identified based on the spinal curvature, muscle imbalance, and body alignment [16,17]. Thus, postural assessments can be performed individually in face-to-face situations.

The number of older adults is on the rise. Therefore, PTs are required to allocate less time per individual in assessing older adults. However, this assessment necessitates the identification of proficient PTs. Consequently, an increased burden on PTs is anticipated. Additionally, the number of missed inspections is increasing, and older adults may be unable to take the appropriate preventive measures. Hence, the development of an automatic posture assessment tool is necessary. This tool is intended to serve as a screening tool prior to inspection. Therefore, we consider the use of deep neural networks (DNNs) for automation.

Currently, DNNs are used to obtain results from a wide range of fields. In particular, image recognition technology using convolutional neural networks (CNNs) has achieved an accuracy equal to or better than that of humans [18]. Considerable studies on image recognition technology, including the inspection of tumors using computed tomography images and screening inspections, such as remote self-checks, have been reported for medical applications [19,20].

The learning methods used in DNNs and CNNs are categorized into supervised, unsupervised, and reinforcement learning [21,22]. This study aimed to develop a supervised CNN model for identifying posture in photographs of older adults. However, crucial teaching data for training have not yet been considered. The teaching data comprise examples of images that form the basis for training and the corresponding correct answer labels. In this posture assessment, the images were either photographs of older adults or those subjected to image processing. The labels correspond to the Kendall classification, ideal or non-ideal for each image. Both the images and labels are pivotal factors in model construction that significantly influence the identification accuracy. Therefore, it is imperative to consider both the photograph itself and the label assignment method before the model construction.

The images in the teaching data should capture the visually identifiable posture of older adults. In the clinic, posture is assessed from the frontal or sagittal plane in various states, such as sitting, standing, or lying [17]. Thus, for photographs, a standing posture from the sagittal plane was deemed suitable, as it allows for visual assessment of the thoracic and lumbar spine curvature and pelvic tilt. However, photographs contain information unrelated to posture, such as clothing, patterns, and backgrounds. Data confirmation and preprocessing must be performed as basic considerations before constructing the model to remove information in the photographs that is irrelevant to posture assessment [23,24]. The use of photographs leads to misrecognition, which interferes with the model accuracy. Additionally, photographs have three channels that are computationally more intensive during training than gray or binary images with a single channel. A reduction in the computational work leads to an increase in the number of middle layers and units, resulting in a redundant model construction. Consequently, two types of images were generated from the captured photographs: a gray image (G image) devoid of red, green, and blue (RGB) color information and a silhouette image (S image) of an older adult. The G images were created to remove the RGB colors from the clothing and background. The S images were created to remove the clothing patterns and boundaries between the clothing and skin of the older adults. Silhouette images have been used as teaching data for deep learning [25], and we consider that S images would also be effective for posture assessment. However, S images contain less information than photographs and lack information on the position of the shoulders and arms; this missing information may be an important feature in posture assessment. Therefore, the posture identified in an S image must be confirmed to be equivalent to the posture assessed in a photograph.

The correct labels for the images in the teaching data and the method for assigning them must be determined. Four labels were planned based on the Kendall classification and two based on ideal and non-ideal postures to provide four or two output layers. It is preferable to have as many teaching data as possible, and the number of labels in the teaching data should be proportional to the number of images. However, the visual and palpation methods (clinical methods) used by PTs are performed under one-on-one face-to-face situations, and it is difficult to secure many labels. As an alternative, we considered label assignment by assessing only the visual method (imaging method). However, the imaging method may assign labels that differ from those of clinical methods. Therefore, differences in the labels between the clinical and imaging methods must be confirmed.

This study aimed to construct a supervised CNN model for posture identification using photographs of older adults. This study evaluated the suitability of S and G images in teaching data and the method of label assignment. From a more advanced perspective, we also proposed some criteria for selecting the teaching images and the possibility of reducing the number of personnel required for majority voting.

## 2. Methods

### 2.1. Stimulus Images and Labels for the Clinical Method

The stimulus image base for the questionnaire survey was obtained during one session at an older adult physical fitness event held in June 2018 and again in June 2019. This session included a postural assessment and standing photographs taken by the PTs. The PTs were 3 PTs with more than 10 years of experience who were routinely involved in postural assessment. The 3 PTs’ assessments were in agreement with each other in the pre-test, so there was considerable validity in these assessments. Thus, these assessments and photographs were used to label the clinical method and as the basis for the stimulus images. The participation of the older adults was voluntary. In total, 658 images were captured (with some duplicates). The photographic data collection was conducted with the approval of the Ethics Committee of the Kyoto Institute of Technology (Protocol Number 2018-19).

The photographs were taken in the sagittal plane, and the entire body of the older adult was captured. The older adults were instructed to position their feet precisely within the marks drawn on the floor for photography purposes. The feet of the participants were placed shoulder-wide apart, and their arms were positioned next to their body. A Kinect v2 color camera was used as the filming device. The camera was fixed on a tripod and positioned 3 m away from the participants.

The assessment was based on the clinical method used by the PTs, and the Kendall classification was categorized into four postures: ideal, KL, SB, and FB. The results of these assessments were used as labels for the clinical methods.

The stimulus images were created using two types of image processing methods. Seven photographs were randomly selected for each of the four postures of the Kendall classification based on the labels of the clinical method. Thus, 28 photographs of 6 Japanese males (mean age, 85 ± 6 years) and 22 Japanese females (mean age, 75 ± 6 years) were processed.

The stimulus images obtained were G and S images of the entire body. Image processing was performed using the following four steps: (1) cropping the entire body of the participant from a photo; (2) grayscale transformation of the cropped photo (G images); (3) extraction of the subjective contours of the whole-body G image; and (4) silhouette processing (S images). The grayscale processing was performed using MATLAB software (R2021b). The subjective contours of the entire body were extracted using Photoshop 2022. After the rim contouring process, the images of the subjects were painted white, and the background was painted black. Figure 1 shows two examples of image processing.

### 2.2. Participants and Ethics

The participants were 28 PTs (mean age: 31.1 ± 8.0 years; mean years of experience: 8.2 ± 6.8 years). All the participants were briefed about the details of the study before participation. These explanations provided an overview of the study, voluntary participation, the protection of personal information, publication and disclosure of the study results, the advantages and disadvantages of study participation, and post-study personal information and data policies. The participants responded to the survey items only after we obtained their consent. This study was reviewed and approved by the Ethics Committee of the Kyoto Institute of Technology (Protocol Number 2022-15).

### 2.3. Data Collection and Items

An online questionnaire survey was conducted in Japan between May and June 2022. The survey web page was constructed using jsPsych version 6.3.1 [26]. The primary item in the survey was an assessment of the Kendall classification based on the stimulus images. The participants selected “ideal”, ”KL”, ”SB”, or ”FB” for the stimulus images presented and a confidence level for their rating from a list of five (1 = possibly, 2 = maybe, 3 = perhaps, 4 = likely, and 5 = probably). The web page for accepting the responses consisted of a display of the stimulus images and check buttons for selecting the Kendall classification categories and confidence levels. The stimulus images were assessed in the order of the S image datasets, followed by the G image datasets. Each dataset contained 28 images that were presented at random. Thus, the participants assessed 56 images. No time limit was set for the responses. Therefore, the participants were allowed to select items until they were satisfied. However, once a response was completed, it could not be changed, and the option to return to the page was prohibited.

### 2.4. Analysis Methods

Before comparing the clinical and imaging methods, it was necessary to determine the ease of assessment of the stimulus images, which were analyzed based on the variability in the responses of the 28 participants. The variability in the responses can be viewed as equivalent to the ambiguity of the information source [27], which was consequently evaluated using the Shannon entropy [28]; based on the four categories, the maximum value of the entropy is 2.0, which is log_2_4. Therefore, when the entropy value was 2.0, the assessments of the 28 participants were the most scattered, whereas the assessments agreed when the entropy was 0. The difference in entropy between all the S and G images was confirmed using the Wilcoxon signed-rank test.

The differences between the images and clinical methods were confirmed according to the different labels assigned to each stimulus image. The labels for the clinical method were determined using the visual and palpation methods used by the PTs. The labels for the imaging method were assigned according to the majority votes of the 28 participants who only identified them using the visual method. When the majority vote was a tie, the label for the imaging method was the Kendall classification category with the highest total confidence level. Thus, three labels were obtained for each stimulus image: the clinical, S, and G imaging methods. Subsequently, a cross-tabulation matrix of the clinical and imaging methods (S and G images) was created from the three labels of the 28 stimulus images. These were created with four categories (ideal, KL, SB, and FB) and two categories, ideal and non-ideal, with the latter combining KL, SB, and FB. Additionally, the agreement was evaluated using the prevalence-adjusted bias-adjusted kappa (PABAK) [29,30], where the PABAK values were defined as follows: <0.00 = no agreement; 0.00–0.20 = slight agreement; 0.21–0.40 = fair agreement; 0.41–0.60 = moderate agreement; 0.61–0.80 = substantial agreement, and 0.81–1.00 = almost perfect agreement [31]. The reliability of the negative and positive results in the clinical and imaging methods was analyzed based on sensitivity and specificity.

The ease of imaging varies greatly depending on the image. Photographs that are difficult to assess can be incorrectly labeled, which reduces the accuracy of the construction of CNN models. Therefore, unstable images must be removed before constructing a model. We considered entropy as a filter for image selection. However, the assumption is that the use of entropy is related to the agreement between the clinical and imaging methods. Therefore, we analyzed the agreement between the entropy values and assessment. The 28 stimulus images were divided into two halves according to their entropy values; the images with the lowest and highest entropy values were defined as low-entropy and high-entropy images, respectively. Cross-tabulation matrices were then created using these images in the same manner as before, and the PABAK values, sensitivity, and specificity were calculated. The differences in the sensitivity and specificity between the low- and high-entropy images were examined using the chi-square test. These calculations were performed using the S and G images.

In this experiment, the labels for the imaging method were assigned based on the majority votes of the 28 participants. The majority of decisions are robustly identifiable as an assessment by a group of PTs, although their years of experience varies from 0 to 25 years. Takizawa et al. reported that differences in the years of experience were identified in the subjective assessment of gait by healthcare professionals in terms of stride width [32]. Therefore, participants with more years of experience may have had different assessments than those with less experience. Participants with more years of experience were also more likely to provide the same ratings for both the clinical and imaging methods. Therefore, we examined the possibility of decreasing the number of participants in the majority group by emphasizing the assessment of participants with more years of experience. Two groups were created based on the years of experience: one with >10 years (12 participants) and the other with <10 years (16 participants) of experience. In these two groups, labeling was conducted again using a majority vote. The agreement between the clinical method and the newly assigned labels was evaluated using a cross-tabulation matrix. The reliability of the negative and positive results was examined based on sensitivity and specificity. In addition, differences in the ratings based on the years of experience were examined based on sensitivity and specificity using the chi-square test. The statistical analyses were performed using SPSS version 28.0 (IBM Corporation, Armonk, NY, USA).

## 3. Results

### 3.1. Variation in the Imaging Method in the Stimulus Images

A box-and-whisker plot of the entropy for the 28 stimulus images to which image processing was applied is shown in Figure 2.

The median values were 1.57 and 1.59, the minimum values were 1.03 and 0.81, and the maximum values were 1.84 and 1.90 for the S and G images, respectively. The quartile ranges were 1.35 to 1.72 for the S images and 1.51 to 1.71 for the G images. The Wilcoxon signed-rank test revealed no significant difference in the entropy values between the image processing methods (Z = 1.345, *p* = 0.178).

### 3.2. Relationship between the Clinical and Imaging Methods

Table 1 presents the cross-tabulation matrices for the clinical and imaging methods, respectively. Table 1 shows the four category assessments based on the Kendall classification and the assessment between ideal and non-ideal.

For the four category assessments in Table 1 based on the clinical method, ideal matched four out of seven S images (57.1%) and five out of seven G images (71.4%); for non-ideal, the KL agreement was <50% for both sets of images (S images, 42.9%; G images, 42.9%), the SB agreement was <50% for both sets of images (S images, 42.9%; G images, 14.3%), and the FB agreement was <50% for both sets of images (S images, 28.6%; G images, 14.3%). The PABAK, a measure of agreement, was −0.14 (*p* = 0.0245, kappa = 0.2381) and −0.29 (*p* = 0.3505, kappa = 0.1429) for the S and G images, respectively.

For the two group assessments, the PABAK was 0.57 (*p* = 0.0233, kappa = 0.4286) and 0.5 (*p* = 0.0228, kappa = 0.4167) for the S and G images, respectively. The sensitivity was 86% for the S images and 76% for the G images, and the specificity was 57% for the S images and 71% for the G images, with sensitivity higher than specificity for both the S and G images. The chi-square test showed no significant difference in sensitivity and specificity (sensitivity, *p* = 0.4319; specificity, *p* = 0.5770).

### 3.3. Agreement of the Assessment in Images Grouped by Entropy

Table 2 shows the cross-tabulation matrix of the clinical and imaging methods by entropy; the PABAK values were higher for the low-entropy images (S images, 0.86, *p* = 0.0034, kappa = 0.7586; G images, 0.86, *p* = 0.0020, kappa = 0.8108) than for the high-entropy images (S images, 0.29, *p* = 0.4805, kappa = 0.1860; G images, 0.14, *p* = 0.5148, kappa = 0.1429).

The sensitivity was higher for the low-entropy images (S images, 100%; G images, 100%) than that for high-entropy images (S images, 70%; G images, 55%). Furthermore, a chi-square test for sensitivity showed a significant difference between the entropy groups for the S and G images (S images, *p* = 0.0497; G images, *p* = 0.0145). The specificity and sensitivity were higher for the low-entropy images (S, 67%; G, 75%) than for the high-entropy images (S, 50%; G, 67%). The chi-square test showed no significant difference in the specificity between the entropy groups (S images, *p* = 0.6592; G images, *p* = 0.8091).

### 3.4. Difference in Assessment by Years of Experience

Table 3 shows the cross-tabulation matrix of the clinical and imaging methods based on years of experience.

The PABAK value was 0.43 (*p* = 0.3927, kappa = 0.1579) for >10 years’ experience and 0.79 (*p* = 0.0001, kappa = 0.7273) for <10 years’ experience for the S images and 0.71 (*p* = 0.0011, kappa = 0.6190) for >10 years’ experience and 0.50 (*p* = 0.0228, kappa = 0.4167) for <10 years’ experience in the G images. The agreement was high for both the S (<10 years) and G (>10 years) images. The sensitivity was higher for the S images (86% for >10 years and 91% for <10 years) than that for the G images (91% for >10 years and 76% for <10 years). The chi-squared test showed no significant difference in the sensitivity between the S and G images (S images, *p* = 0.6337; G images, *p* = 0.2141). The specificity was high for both the S (29% for >10 years; 86% for <10 years) and G (76% for >10 years; 71% for <10 years) images but not for the S images for >10 years’ experience. The chi-square test showed a significant difference in specificity for the S images (*p* = 0.0307) but not for the G images (*p* = 1.0000).

## 4. Discussion

### 4.1. Evaluating the Variation in the Responses in the Imaging Method

In this study, we tested the ease of assessing the stimulus images. The entropy analysis revealed that the assessment of these images included a wide range, from concentrated to dispersed, and many stimulus images showed entropy values >1, half of the maximum value of 2. Therefore, identification of posture using the imaging method was considered difficult overall, although several images exhibited entropy values of <1. This suggests that several postures can be identified, even when using the imaging method.

The difference between the assessments of the two types of images was confirmed using the Wilcoxon’s signed-rank test. The results showed no difference in entropy between the S and G images. Kawamoto reported a moderate correlation between the impressions of two clothing images and their silhouettes [33]. In the S images in the present study, it is thought that the participants could estimate to a certain extent the basic check points for postural assessment, such as head and neck flexion, spinal column, and pelvic kyphosis [34], from the contours through visual completion. Therefore, although variability was present in the assessment, it was not considered to be caused by the image processing.

### 4.2. Investigating the Agreement of the Assessment between Clinical and Imaging Methods

In this study, the differences between the clinical and imaging methods of the four and two categories were tested using a cross-tabulation matrix. The PABAK values in the four categories were low, and no concordance was found when Landis’ criteria were applied. Focusing on each Kendall classification, the ideal category showed high agreement in >50% of the cases, with five of seven images in agreement. However, the KL, SB, and FB categories of the non-ideal group showed low agreement (<50%), suggesting that the low value of PABAK was influenced by the discrepancy in poor posture. These results indicate that ideal posture can be effectively assessed using the imaging method, as can the clinical method. However, the non-ideal category suggested that neither KL, SB, nor FB could be identified using the imaging method. The Kendall classification is not only based on the skeleton and position of each landmark but also considers muscle imbalance as an important identifying factor [12,15]. Yamada et al. reported that distinguishing between proper pelvic angles and appearance is difficult [16]. In the imaging method, the subjects could not determine the exact position of each landmark and muscle imbalance, which may have made it difficult to classify poor posture in detail. Differences due to the image processing also had no effect, and when half of the subjects were used as references, agreement was observed for the ideal but not for the non-ideal categories. Furthermore, assessment has been shown to be difficult due to variations in entropy. These results suggest that it is difficult to classify these four categories using an imaging method.

For the ideal and non-ideal categories, the PABAK value was 0.57 for the S images and 0.50 for the G images; after applying Landis’ criteria, both the S and G images showed moderate agreement. The agreement was improved by the integration of poor posture. Furthermore, the precision of the imaging method was confirmed by the sensitivity and specificity, with high sensitivity values of 86% and 76% for the S and G images, respectively. High sensitivity indicates the correct detection of non-ideal posture, and this can be assessed using the imaging method. The specificity was 57% for the S images and 71% for the G images, which was lower than the sensitivity. In the case of the G images, ideal posture was detected approximately 70% of the time, even using the imaging method. Because sensitivity and specificity show a reciprocal relationship, it is necessary to determine the balance between them depending on the content of the inspection [35]. For the G images, the sensitivity and specificity were both approximately 70%; therefore, the precision was unbiased and balanced. Thus, for the two categories, identification using the imaging method was reliable for both the ideal and non-ideal postures. Accordingly, the labels for the teaching data can be assigned using the imaging method. However, the S images had a slightly lower specificity and were significantly different from the G images, based on the chi-square test. Because this study aimed to construct a CNN for screening inspection, sensitivity was more important than specificity. Therefore, although the specificity differed between the S and G images, the S images could be used as teaching data for the CNN.

### 4.3. Suggesting Indicators for Teaching Data

The 28 stimulus images were divided into low- and high-entropy images, and the agreement between the two was evaluated using the PABAK values. The low-entropy images had larger PABAK values than the 28 stimulus images for both the S and G images. By applying Landis’ criteria, the PABAK values of the low-entropy images were found to be almost consistent. However, the high-entropy images were smaller than the 28 stimulus images for both the S and G images, and no agreement was found in Landis’ criteria. These results suggest that entropy is related to the agreement between the clinical and imaging methods. Therefore, the variability in the responses using the imaging method can be regarded an indicator of the correctness of the imaging method compared to the clinical method. Because the images of the teaching data in deep learning are directly related to the accuracy of the model, image selection is desired during preprocessing [23,24]. We believe that the measure of entropy presented here can be used to assess the availability of the images.

Low-entropy images have a higher sensitivity and specificity than high-entropy images. The sensitivity was significantly different based on the chi-square test, whereas the specificity was not. As the purpose of the constructed model is screening inspection, the difference in sensitivity is an important factor in ensuring reliability. We considered that entropy could be used to select the images for teaching data.

This study compared the sensitivity and specificity of the PABAK between two groups of participants with more or less than 10 years of experience. Those with >10 years’ experience had higher PABAK scores, sensitivity, and specificity when using the G images than those with <10 years’ experience. The PABAK was 0.79 for the participants with >10 years’ experience and 0.50 for those with <10 years’ experience. By applying Landis’ criteria, >10 years’ experience can be considered a moderate agreement and <10 years’ experience a fair agreement. Neither sensitivity nor specificity showed significant differences. Matsunaga stated that the effect of differences in years of clinical experience on the thinking process is that participants with more years of experience base their evaluations on their own acquired experience rather than on the manual criteria [36]. It was difficult to assess the stimulus images targeted in this study using entropy. Thus, participants’ own experiential knowledge is important in addition to basic knowledge; the participants with <10 years of experience showed differences in posture identification because they had fewer years of experience. The posture identification by the participants with >10 years of experience was consistent with the clinical method, and it affected the PABAK values. Thus, it is possible to reduce the number of participants who assign labels to the teaching data when 10 years of experience is used as the criterion. However, in the S images, the PABAK values were higher for those with <10 years of experience. In particular, the specificity was significantly lower for the participants with >10 years of experience, and a significant difference was observed between them and those with <10 years of experience. We consider that low specificity values were produced because the ideal category was not selected by them as often as it was by the other participants. The participants complemented each landmark with the body contours to identify the posture. Because visual completion is related to the participant’s own amount of knowledge [37,38], those with >10 years’ experience are likely to infer more factors of poor posture with greater visual completion than those with <10 years’ experience. Therefore, we consider ideal posture to almost never be selected. Accordingly, the most informative and explicit G images should be used to label the teaching data.

### 4.4. Limitations

Furthermore, it is necessary to discuss the limitations of the sample size. In this study, each Kendall classification category was randomly allocated seven photographs. Although the number of images assessed was small due to the field experiment, the photographs were taken in a clinical setting, encompassing a broad spectrum of images. These ranged from easily assessable to challenging. Moreover, considering the burden of assessing PTs, it is necessary to maintain the number of photographs to a certain extent. Therefore, we considered a sample size of 28 pictures, including a small number of pictures and large number of acceptable postures. However, due to the issues of type I and II errors, an increase in the number of photographs should be considered to improve the reliability of the model in the future. In particular, the number of photographs used in this experiment is smaller than ideal, and it is quite likely that the specificity was affected by type I or II errors. Therefore, the number of photographs calls for a further increase with an emphasis on ideal posture. Moreover, the sample exhibited a gender imbalance, with 6 males and 22 females. Consequently, the potential influence of sex differences on the sample images warrants further consideration. Gender differences in the S images can be confirmed by the contours. However, the agreement between the clinical and imaging methods for the two categories was 83% for males and 73% for females for both the S and G images. Although the number of male participants was small, the influence of sex on the assessment was considered to be small.

Furthermore, the limitations of PT assessment using imaging methods must be discussed. A preliminary experiment was conducted to discern the differences between the S and G images using five PTs. In the case of the two categories, the PTs provided the same assessment without any differences in the image processing. The posture of each image was determined according to a majority vote of 28 participants to ensure the robustness of the assessment.

Furthermore, the limitations of the S and G images must be discussed. The images present issues, such as per-pixel noise and high computational costs. Therefore, although S and G images can be used, we consider that additional processing, such as resizing, is necessary for model construction.

Furthermore, the various clothing and body shapes of the older adults in the images must be discussed. In the case of large-sized clothing, the contours of an older adult’s body cannot be recognized in the image. Unclear contours can cause mislearning when convolving images. The same may be said of characteristic body shapes. Therefore, it is necessary to remove these images for the accuracy of the model construction. However, we consider that these problems can be addressed by subdividing the number of output layers as the number of teaching data increases.

Furthermore, differences between other technologies and imaging methods must be discussed. There are a wide range of posture assessments, from contact, in which sensors are attached to the joints to acquire data, to non-contact, in which optical sensors or cameras are used. Non-contact methods use optical or depth sensors to determine the shape of an object. However, the imaging method is applicable to non-contact conditions and provides less information than other technologies. However, the ease of data acquisition is a significant advantage, and it does not require specialized knowledge for its use. Additionally, there is technology for the virtual construction of three-dimensional humanoid models that combines depth cameras and markerless motion capture [39,40]. Although this technology has the advantage of a high information content, it requires expertise in data acquisition. However, data acquisition using the imaging method was simple. This method has two limitations, making it difficult to identify the category of non-ideal. In contrast, the S and G images are highly convenient because they can provide additional information, such as the location of key points based on skeletal estimation. Therefore, there is a possibility that three or four categories can be identified depending on the additional information and that even PTs with less experience may be able to assess the results to the same degree as using the clinical method. As a large number of images and labels will be required to construct CNN models in the future, we believe that the imaging method can be used to collect them while maintaining reliability. In addition to model construction, the imaging method can also be used for preliminary inspection in the clinic and will lead to a reduction in the time spent on the inspection of each individual patient, early detection of symptoms, and integration into other fields, such as rehabilitation and prevention. Furthermore, assessment can be performed without a face-to-face situation. However, as the teaching data in this proposal are based on older adults, it is necessary to consider a wide range of age groups, such as children and adolescents. Furthermore, considering the long-term operation of the model after its construction, it is necessary to add a system that not only identifies but also proposes specific preventive measures to avoid poor posture. In connection with this, we consider that the imaging method is a valid method for labeling large numbers of teaching data.

## 5. Conclusions

In this study, we examined the images and labels and their determination methods in teaching data prior to CNN construction. We also investigated the reduction in the number of personnel involved in selecting and labeling images based on entropy.

Assessment of the S and G images showed no difference based on the PABAK and sensitivity. However, a significant difference was observed in specificity. Despite this, S images can be used as teaching data when screening inspections are performed.

For ideal and non-ideal postures, a moderate agreement was present between the clinical and imaging methods in terms of the label assignment. The sensitivity and specificity of the G images were both >70%, suggesting that the imaging method could be used to perform the same assessment as the clinical method on G images.

The entropy of the variability in the assessment was related to the agreement between the clinical and imaging methods. Therefore, the entropy value may be an indicator for the selection of images for teaching data.

A difference between the participants with >10 years of experience and those with <10 years of experience was confirmed. The participants with >10 years of experience were more consistent with the clinical method than those with <10 years of experience. Therefore, the number of participants could be reduced based on whether they had 10 years of experience in assigning labels to the teaching data.

## Figures and Tables

**Figure 1 geriatrics-09-00040-f001:**
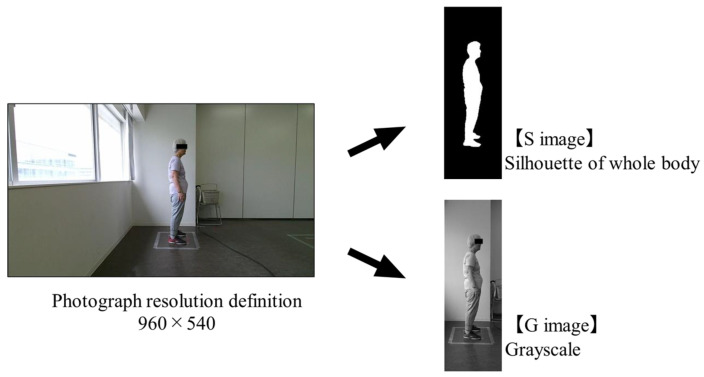
Photograph and image processing examples.

**Figure 2 geriatrics-09-00040-f002:**
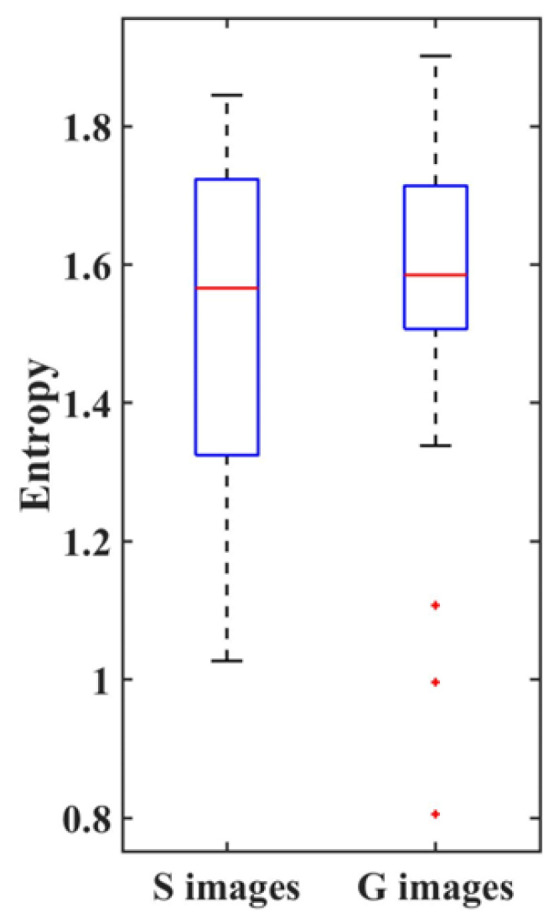
Box-and-whisker plot of entropy for the stimulus images.

**Table 1 geriatrics-09-00040-t001:** Cross-tabulation matrices of the clinical and imaging methods.

			Imaging method								
			S images					G images			
			Ideal	Non-ideal				Ideal	Non-ideal		
				KL	SB	FB			KL	SB	FB
Clinical method	Ideal		4	0	0	3		5	1	0	1
			**4**	**3**				**5**	**2**		
		KL	0	3	0	4		0	3	2	2
	Non-ideal	SB	0	2	3	2		2	3	1	1
		FB	3	0	2	2		3	1	2	1
			**3**	**18**				**5**	**16**		

Bold text in the table indicates totals by color.

**Table 2 geriatrics-09-00040-t002:** Cross-tabulation matrices of clinical and imaging methods by high and low entropy.

		Imaging method										
		S images						G images				
		Low			High			Low			High	
		Ideal	Non-ideal		Ideal	Non-ideal		Ideal	Non-ideal		Ideal	Non-ideal
Clinical method	Ideal	2	1		2	2		3	1		2	1
	Non-ideal	0	11		3	7		0	10		5	6

**Table 3 geriatrics-09-00040-t003:** Cross-tabulation matrices of clinical and imaging methods by years of experience.

		Imaging method										
		S images						G images				
		<10 years			>10 years			<10 years			>10 years	
		Ideal	Non-ideal		Ideal	Non-ideal		Ideal	Non-ideal		Ideal	Non-ideal
Clinical method	Ideal	6	1		2	5		5	2		5	2
	Non-ideal	2	19		3	18		5	16		2	19

## Data Availability

The data are available from the corresponding author upon request.

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
