# Peer review of "Agreement in the Postural Assessment of Older Adults by Physical Therapists Using Clinical and Imaging Methods"

_geriatrics, 2024, doi:10.3390/geriatrics9020040_

Round 1

Reviewer 1 Report

Comments and Suggestions for Authors

Comments to the Author

The authors illustrated a very valuable issue that would provide basic evidence for making the postural assessment for older adults simpler and automated. However, there are a couple of additional comments on the manuscript.

1. The authors declared “Moreover, considering the burden of assessing PTs, it is necessary to maintain the number of photographs to a certain extent. Therefore, we considered a sample size of 28 pictures, including a small number of pictures and large number of postures acceptable”. In sample size determination, the burden, type I error and power should be consideration. Obviously, the authors missed the type I error and power when determined the sample size.

2. The stimulus image was obtained during one session of an older adult physical fitness event, which included a postural assessment and standing photographs. However, the accuracy and the PTS’ background information was unclear. It is meaningless to assess the agreement between the clinical method and the image method when the PTS’ the were un-comparable.

3. “Table 1” and “Table 2” should be compiled into one table.

4. There should be P-value corresponding to the PABAK values.

5. Chis-square values and the corresponding P-values should be displayed in the revised manuscript.

Author Response

Date: 14th March, 2024

To:  Reviewer #1,

From:  Naoki Sugiyama

Re.: 

Title:  Agreement in the postural assessment of older adults by physical therapists using clinical and imaging methods

Thank you for your revision invitation letter with reviewer’s comments.

Point-by-point replies to the comments are presented in this letter.

The red-letter sentences were changed in the revised manuscript.

We hope this revision will be satisfactory for the Reviewers as well as your editorial team, and our paper will be accepted for publication in the Journal of Geriatrics.

We look forward to hearing from you at your earliest convenience.

Sincerely yours,

Naoki Sugiyama,

Doctoral Programs of Department of Advanced Fibro-Science, Kyoto Institute of Technology, Kyoto, Japan

Hashikami-cho, Matsugasaki, Sakyo-ku, Kyoto 606-8585, Japan.

Tel: +81-75-724-7014, Email: nsugiyama.u@gmail.com

---

Reply to the comments from Reviewer #1

  1. The authors declared “Moreover, considering the burden of assessing PTs, it is necessary to maintain the number of photographs to a certain extent. Therefore, we considered a sample size of 28 pictures, including a small number of pictures and large number of postures acceptable”. In sample size determination, the burden, type I error and power should be consideration. Obviously, the authors missed the type I error and power when determined the sample size.

We have added the type I and II error (Page 10, Lines 423-425).

However, as the issues of type I and II error, an increase in the number of photographs should be considered to improve the reliability of the model in the future.

  1. The stimulus image was obtained during one session of an older adult physical fitness event, which included a postural assessment and standing photographs. However, the accuracy and the PTS’ background information was unclear. It is meaningless to assess the agreement between the clinical method and the image method when the PTS’ the were un-comparable.

We have added the PTs of an older adult physical fitness event (Page 3, Lines 140-143).

The PTs were 3 PTs with more than 10 years of experience who were routinely involved in postural assessment. 3 PTs' assessments were in agreement with each other in the pre-test, there is considerable validity in these assessments.

  1. “Table 1” and “Table 2” should be compiled into one table.

We have revised Tables (Page 6-7, Lines 268-269).

  1. There should be P-value corresponding to the PABAK values.
  2. Chis-square values and the corresponding P-values should be displayed in the revised manuscript.

We have revised and added the P-value (Page 7-8, Lines 276-313).

・The PABAK, a measure of agreement, was -0.14 (p = 0.0245) and -0.29 (p = 0.3505) for the S and G images, respectively.

・For the two group assessments, the PABAK was 0.57 (p = 0.0233) and 0.5 (p = 0.0228) for the S and G images, respectively.

・The chi-square test showed a significant difference in specificity (sensitivity, p = 0.4319; specificity, p = 0.0075).

・Table 2 shows the cross-tabulation matrix of clinical and imaging methods by entropy; PABAK values were higher for low-entropy images (S images, 0.86, p = 0.0034; G images, 0.86, p = 0.0020) than for high-entropy images (S images, 0.29, p = 0.4805; G images, 0.14, p = 0.4028).

・Furthermore, a chi-square test for sensitivity showed a significant difference between the entropy groups for the S and G images (S images, p = 0.0497; G images, p = 0.0145).

・The chi-square test showed no significant difference in specificity between the entropy groups (S images, p = 0.6592; G images, p = 0.8091).

・The PABAK value was 0.43 (p = 0.3927) for >10 years and 0.79 (p = 0.0001) for <10 years for the S images, and 0.71 (p = 0.0011) for >10 years and 0.50 (p = 0.0228) for <10 years in the G images.

・The chi-squared test showed no significant difference in sensitivity between the S and G images (S images, p = 0.6337; G images, p = 0.2141).

・The chi-square test showed a significant difference in specificity for the S images (p = 0.0307) but not for the G images (p = 1.0000).

Reviewer 2 Report

Comments and Suggestions for Authors

Dear Authors, I appreciate your research. Incorporating AI into physical therapy practice is an important topic and a useful practice. Your research will contribute to improving this process. My only doubts concern the work restrictions that you did not write about in the paper. You should pay attention to 1. the patient's clothing (is it too baggy, loose or are there too many layers), 2, what if the patient is obese? Your proposal is very suitable as a first screening test and should be treated as such (as you emphasized). The manuscript is written in an appropriate language and contains all the necessary elements.  I wish you further scientific success

Author Response

Date: 14th March, 2024

To:  Reviewer #2,

From:  Naoki Sugiyama

Re.: 

Title:  Agreement in the postural assessment of older adults by physical therapists using clinical and imaging methods

Thank you for your revision invitation letter with reviewer’s comments.

Point-by-point replies to the comments are presented in this letter.

The red-letter sentences were changed in the revised manuscript.

We hope this revision will be satisfactory for the Reviewers as well as your editorial team, and our paper will be accepted for publication in the Journal of Geriatrics.

We look forward to hearing from you at your earliest convenience.

Sincerely yours

Naoki Sugiyama,

Doctoral Programs of Department of Advanced Fibro-Science, Kyoto Institute of Technology, Kyoto, Japan

Hashikami-cho, Matsugasaki, Sakyo-ku, Kyoto 606-8585, Japan.

Tel: +81-75-724-7014, Email: nsugiyama.u@gmail.com

---

Reply to the comments from Reviewer #2

  1. The patient's clothing (is it too baggy, loose or are there too many layers)
  2. what if the patient is obese?

We have added the clothing and body shapes (Page 10-11, Lines 442-448).

Furthermore, the various clothing and body shapes older adults in image must be discussed. In the case of size of clothing large, the contours of older adult's body cannot be recognized in the image. Unclear contours can cause mislearning when convolving images. The same may be said of characteristic body shapes. Therefore, it is necessary to re-move these images for accuracy of model construction. However, we consider that these problems can be addressed by subdividing the number of output layers as the number of teaching data increases.

Reviewer 3 Report

Comments and Suggestions for Authors

The study investigates the agreement between clinical and imaging methods for postural assessment in older adults, utilizing grayscale and silhouette images alongside traditional methods. It aims to determine if imaging can reliably substitute or augment clinical assessments. The findings demonstrate the potential of imaging methods in posture assessment, although challenges remain in classifying non-ideal postures. The study highlights the importance of image selection and proposes using entropy as a filter to enhance model accuracy.

Based on the content overview provided, here are further review suggestions:

  1. Increase the number of Participants: It is recommended to collect more samples for analysis, which will make the research results more convincing.
  2. Incorporate Comparative Analysis: Compare the proposed imaging methods with current gold-standard clinical assessments to highlight advantages or limitations more clearly.
  3. What are the advantages of using imaging methods for postural assessment over direct clinical assessment by physical therapists? How can imaging methods for postural assessment be used to supplement or enhance traditional assessment methods?
  4. Future Research Directions: Suggest specific future research directions, such as the integration of advanced machine learning techniques for better posture classification or exploring the applicability of this method to other physical conditions.

These suggestions aim to strengthen the manuscript by enhancing its providing a clear path for future research and development in the field for postural assessment in older adults.

Author Response

Date: 14th March, 2024

To:  Reviewer #3,

From:  Naoki Sugiyama

Re.: 

Title:  Agreement in the postural assessment of older adults by physical therapists using clinical and imaging methods

Thank you for your revision invitation letter with reviewer’s comments.

Point-by-point replies to the comments are presented in this letter.

The red-letter sentences were changed in the revised manuscript.

We hope this revision will be satisfactory for the Reviewers as well as your editorial team, and our paper will be accepted for publication in the Journal of Geriatrics.

We look forward to hearing from you at your earliest convenience.

Sincerely yours,

Naoki Sugiyama,

Doctoral Programs of Department of Advanced Fibro-Science, Kyoto Institute of Technology, Kyoto, Japan

Hashikami-cho, Matsugasaki, Sakyo-ku, Kyoto 606-8585, Japan.

Tel: +81-75-724-7014, Email: nsugiyama.u@gmail.com

---

Reply to the comments from Reviewer #3

  1. Increase the number of Participants: It is recommended to collect more samples for analysis, which will make the research results more convincing.

We have added the limitation of sample collection (Page 10, Lines 423-425).

However, as the issues of type I and II error, an increase in the number of photographs should be considered to improve the reliability of the model in the future.

  1. Incorporate Comparative Analysis: Compare the proposed imaging methods with current gold-standard clinical assessments to highlight advantages or limitations more clearly.

We have revised and added the limitation of imaging method (Page 11, Lines 459-460).

This method has two limitations, making it difficult to identify the category of non-ideal.

  1. What are the advantages of using imaging methods for postural assessment over direct clinical assessment by physical therapists? How can imaging methods for postural assessment be used to supplement or enhance traditional assessment methods?

We have revised and added the advantage of imaging method (Page 11, Lines 467-471).

In addition to model construction, the imaging method can also be used for preliminary inspection in the clinic, and will lead to a reduction in the time spent on inspection of each individual patient, early detection of symptoms, and integration into other fields, such as rehabilitation and prevention. Furthermore, assessment can be performed without a face-to-face situation.

  1. Future Research Directions: Suggest specific future research directions, such as the integration of advanced machine learning techniques for better posture classification or exploring the applicability of this method to other physical conditions.

We have revised and added the future research (Page 11, Lines 471-477).

However, as the teaching data in this proposal are based on older adults, it is necessary to consider a wide range of age groups, such as children and adolescents. Furthermore, considering the long-term operation of the model after its construction, it is necessary to add a system that not only identifies but also proposes specific preventive measures to avoid poor posture. In this connection, we consider that the imaging method is a valid method for labeling large amounts of teaching data.

Round 2

Reviewer 1 Report

Comments and Suggestions for Authors

The authors replied all the comments. I am sorry that there are additional comments.

1.     The Chi-square test may inappropriate for the sample size less than 40. It may enlarge the Type I error or Type II error, especially for the comparison of specificity for the S images between the experience group.

2.     The Kappa index should be displayed along with the PABAK.

3. The P-value(0.4028) corresponding to the PABAK for the high-entropy G images may be mistake. Please check it.

Author Response

Date: 19th March, 2024

To:  Reviewer #1,

From:  Naoki Sugiyama

Re.: 

Title:  Agreement in the postural assessment of older adults by physical therapists using clinical and imaging methods

Thank you for your revision invitation letter with reviewer’s comments.

Point-by-point replies to the comments are presented in this letter.

The blue-letter sentences were changed in the revised manuscript.

We hope this revision will be satisfactory for the Reviewers as well as your editorial team, and our paper will be accepted for publication in the Journal of Geriatrics.

We look forward to hearing from you at your earliest convenience.

Sincerely yours,

Naoki Sugiyama,

Doctoral Programs of Department of Advanced Fibro-Science, Kyoto Institute of Technology, Kyoto, Japan

Hashikami-cho, Matsugasaki, Sakyo-ku, Kyoto 606-8585, Japan.

Tel: +81-75-724-7014, Email: nsugiyama.u@gmail.com

---

Reply to the comments from Reviewer #1

  1. The Chi-square test may inappropriate for the sample size less than 40. It may enlarge the Type I error or Type II error, especially for the comparison of specificity for the S images between the experience group.

We have added the type I and II error in the specificity (Page 10, Lines 428-430).

In particular, the number of photographs used in this experiment is small for ideal, and it is quite likely that the specificity was affected by type I or II error. Therefore, the number of photographs calls for further increase with an emphasis on ideal.

  1. The Kappa index should be displayed along with the PABAK.

We have added the Kappa (Page 7-8, Lines 276-307).

・The PABAK, a measure of agreement, was -0.14 (p = 0.0245, kappa = 0.2381) and -0.29 (p = 0.3505, kappa = 0.1429) for the S and G images, respectively.

・For the two group assessments, the PABAK was 0.57 (p = 0.0233, kappa = 0.4286) and 0.5 (p = 0.0228, kappa = 0.4167) for the S and G images, respectively.

・PABAK values were higher for low-entropy images (S images, 0.86, p = 0.0034, kappa = 0.7586; G images, 0.86, p = 0.0020, kappa = 0.8108) than for high-entropy images (S images, 0.29, p = 0.4805, kappa = 0.1860; G images, 0.14, p = 0.5148, kappa = 0.1429).

The PABAK value was 0.43 (p = 0.3927, kappa = 0.1579) for >10 years and 0.79 (p = 0.0001, kappa = 0.7273) for <10 years for the S images, and 0.71 (p = 0.0011, kappa = 0.6190) for >10 years and 0.50 (p = 0.0228, kappa = 0.4167) for <10 years in the G images.

  1. The P-value(0.4028) corresponding to the PABAK for the high-entropy G images may be mistake. Please check it.

We have revised the P-value (Page 7, Line 288).

(S images, 0.29, p = 0.4805, kappa = 0.1860; G images, 0.14, p = 0.5148, kappa = 0.1429).

Moreover, we have revised the P-value (Page 7, Lines 282-283).

The chi-square test showed no significant difference in sensitivity and specificity (sensitivity, p = 0.4319; specificity, p = 0.5770).